 **eLIFE**

# Skill learning strengthens cortical representations of motor sequences

**Tobias Wiestler, Jörn Diedrichsen***

Institute of Cognitive Neuroscience, University College London, London, United Kingdom

**Abstract** Motor-skill learning can be accompanied by both increases and decreases in brain activity. Increases may indicate neural recruitment, while decreases may imply that a region became unimportant or developed a more efficient representation of the skill. These overlapping mechanisms make interpreting learning-related changes of spatially averaged activity difficult. Here we show that motor-skill acquisition is associated with the emergence of highly distinguishable activity patterns for trained movement sequences, in the absence of average activity increases. During functional magnetic resonance imaging, participants produced either four trained or four untrained finger sequences. Using multivariate pattern analysis, both untrained and trained sequences could be discriminated in primary and secondary motor areas. However, trained sequences were classified more reliably, especially in the supplementary motor area. Our results indicate skill learning leads to the development of specialized neuronal circuits, which allow the execution of fast and accurate sequential movements without average increases in brain activity.

## Introduction

The human brain has a remarkable ability to learn complex motor skills. However, the neural changes that underlie this ability remain largely unknown. Although functional magnetic resonance imaging (fMRI) studies have shown that learning can lead to both activity increases and decreases in primary and secondary motor areas (for reviews, see *Dayan and Cohen, 2011*; *Penhune and Steele, 2012*; *Hardwick et al., 2013*), these average activity changes remain hard to interpret. On the one hand, motor skill acquisition may lead to increased neural recruitment for trained behaviors, thereby increasing average activation (*Grafton et al., 1995*; *Karni et al., 1995*; *Hazeltine et al., 1997*; *Floyer-Lea and Matthews, 2005*; *Lehericy et al., 2005*; *Penhune and Doyon, 2002*; *Penhune and Doyon, 2005*). Alternatively, learning may result in the development of representations that produce the same behavior with higher neural efficiency, thereby reducing activity (*Jenkins et al., 1994*; *Toni et al., 1998*; *Ungerleider et al., 2002*; *Poldrack et al., 2005*; *Xiong et al., 2009*; *Ma et al., 2010*; *Penhune and Doyon, 2005*). Finally, motor practice may induce simultaneous signal increases (due to increased neural recruitment) and signal decreases (due to more efficient encoding), making motor learning difficult to detect using traditional fMRI paradigms (*Steele and Penhune, 2010*).

Here we test the idea that the learning of fast motor sequences, a commonly-used task in the study of skill learning, causes the development of more distinct cortical activation patterns associated with each individual sequence, independent of average activity changes. Consider the neural activation patterns for two different finger sequences in areas with different types of movement representations (*Figure 1A–C*). A region that controls elementary hand movements independent of their sequential context may comprise five separate neuronal populations (*Figure 1A*), each of which a dynamical system (*Churchland et al., 2012*), whose activation causes the production of an individual finger press. Because each sequence contains each finger once, the two sequences will activate the same cortical patches, albeit in a different temporal order. Due to the low temporal resolution of fMRI, in particularly when averaging, as done here, over three executions of the same sequence, the two sequences would be therefore associated with identical activity patterns.

***For correspondence:**
j.diedrichsen@ucl.ac.uk

**Competing interests:** The authors declare that no competing interests exist.

**Reviewing editor**: Jody C Culham, University of Western Ontario, Canada

**eLife digest** Functional magnetic resonance imaging (fMRI) is a widely used technique that makes it possible to observe changes in a person's brain activity as they perform specific tasks while lying in a scanner. These could range from listening to music or looking at images, to recalling words or imagining a scene, and each will produce a distinct pattern of neural activity.

However, fMRI data can be difficult to interpret. Say a particular area of the brain is very active when a subject is trying to perform a new task, but becomes less active as the subject becomes better at the task and performs it more easily. Does this mean that the brain region is used for learning the task, but not for performing once it has been learned? Or alternatively, does it show that the brain area is involved in carrying out the task, but that it becomes more efficient with practice, and so shows less activity in later scans?

Now, Wiestler and Diedrichsen have obtained data that help to distinguish between these alternatives. Subjects were trained to carry out four specific sequences of finger movements and then asked either to reproduce these 'trained' sequences or to perform four 'untrained' sequences while in the fMRI scanner. All eight sequences produced high levels of activity in the areas of motor cortex that control finger movements.

However, closer analysis showed marked differences between the patterns of activity produced during the 'trained' sequences and those seen during 'untrained' sequences that involved moving the same fingers.

Wiestler and Diedrichsen proposed that when subjects train to perform specific movement sequences, this should lead to the development of neural circuits that are specialized to carry out those specific movements—and that detailed analysis of the fMRI data would allow them to identify patterns of activity that correspond to these circuits. Sure enough, when they analysed the fMRI scans, Wiestler and Diedrichsen found that the activation patterns associated with 'trained' movement sequences were more readily distinguishable from each other than those associated with the 'untrained' movement sequences, even in areas where training led to an overall reduction in activity.

As well as showing that movement sequences become associated with specific spatial patterns of activation as they are learned, this study provides a new way to study learning in fMRI that should be useful for many future studies.

Single cell recording studies have demonstrated that supplementary motor area (SMA; *Tanji and Shima, 1994*) and primary motor cortex (M1; *Matsuzaka et al., 2007*) are sensitive to the sequential context and that individual neurons are preferentially active for transitions between specific movements. The development of such neuronal populations with training is thought to underlie the faster production of trained sequences (*Matsuzaka et al., 2007*). Importantly, the existence of such units would cause the two sequences to rely on partly separate neuronal populations, that is they should be associated with slightly different activity patterns (*Figure 1B*). After prolonged training, some neurons even appear to code for longer fragments, firing preferentially at the beginning of specific sequences (*Tanji and Shima, 1994*). Areas containing many of such units would therefore activate even more distinct neuronal populations for particular trained sequences (*Figure 1C*). This idea therefore predicts that trained sequence should be associated with the activation of more distinct neuronal populations than comparable untrained sequences.

If these neuronal populations differ sufficiently in the spatial distribution of their presynaptic activity, then each sequence should be associated with a distinct spatial activity pattern that may be detectable using fMRI. We tested this idea using 'multi-voxel pattern analysis' (MVPA), which detects differences in voxel-by-voxel activity patterns in an area of cortex, even if these patterns are highly overlapping and defined by neuronal differences on a small spatial scale (*Kamitani and Tong, 2005*; *Swisher et al., 2010*; *Freeman et al., 2011*). Participants were trained for 4 days to produce four different movement sequences with their left hand. After training, participants underwent two fMRI scans, performing either four trained or four untrained sequences. Using MVPA, we tested whether activity patterns of trained sequences could be discriminated from each other more easily than could untrained sequences, and how the representational structure changes with learning.

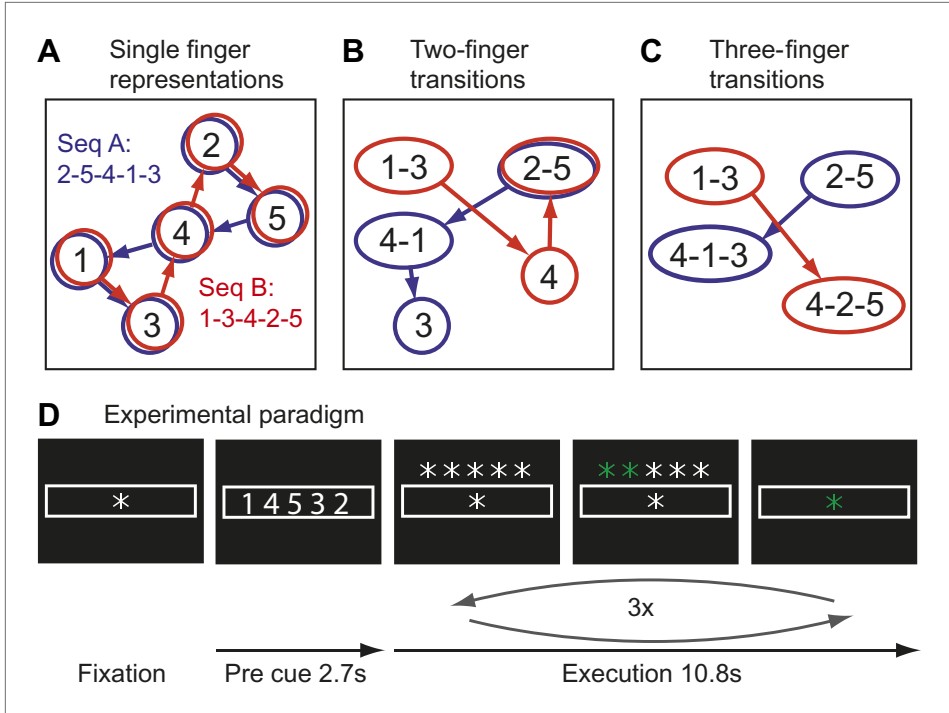

Figure 1. Hypothetical learning-related changes in activity patterns associated with two sequences of five finger-presses. (**A**) If a region consists of units that are preferentially activated for single finger-presses, both sequence A (blue) and sequence B (red) activate the same units in a different temporal order. (**B**) As a region develops units that preferentially encode specific finger transitions, sequence A and B will activate partly separate and partly overlapping components of the network. (**C**) In a network that is highly specialized for specific multiple-finger transitions, the two sequences activate independent parts of the network. (**D**) In the experimental paradigm, a sequence was cued, and then executed three times from memory; fMRI activity was averaged over instruction and execution phase.

## Results

### Behavioral correlates of skill learning

Movement sequences were instructed using a string of five numbers, indicating from left to right the digits to be pressed with one referring to the thumb and five to the little finger (*Figure 1D*). Each sequence consisted of the same five isometric finger presses in a different order. Participants memorized the sequence and then executed it once as fast as possible triggered by a go-cue. This execution was repeated either 5 (training) or 3 times (scanning).

Initially, participants executed the sequences slowly and deliberately with pauses between individual presses (*Figure 2A*). After 4 days of training on four selected sequences with the left hand, movement times (MTs; *Figure 2C*, blue line) reduced by approximately half. At the end of training individual finger presses overlapped considerably (*Figure 2B*).

To assess generalization, participants were probed before and after training on both the trained sequences and on four novel, untrained sequences. The results show that one aspect of the acquired skill was sequence-specific: the left-hand MTs were 237 ms (SE: ±42 ms) faster for trained than for untrained sequences. This difference was highly significant, also when correcting for small difference between the sequences at pre-test, $t(14) = 5.749$, $p<0.0001$. MTs for the untrained sequences dropped by 796 ms (±97 ms) compared to pre-test, $t(15) = 8.23$, $p<0.001$, suggesting that a considerable part of the acquired skill was general. For example, participants may have become faster at movement transitions between specific finger pairs. This ability would also enhance the production of untrained sequences, which shared 59.6% of digit transitions with the trained sequences. Therefore, the comparison of neural representations of trained and untrained sequences after learning will reveal correlates of the sequence-specific rather than general skill acquired during learning.

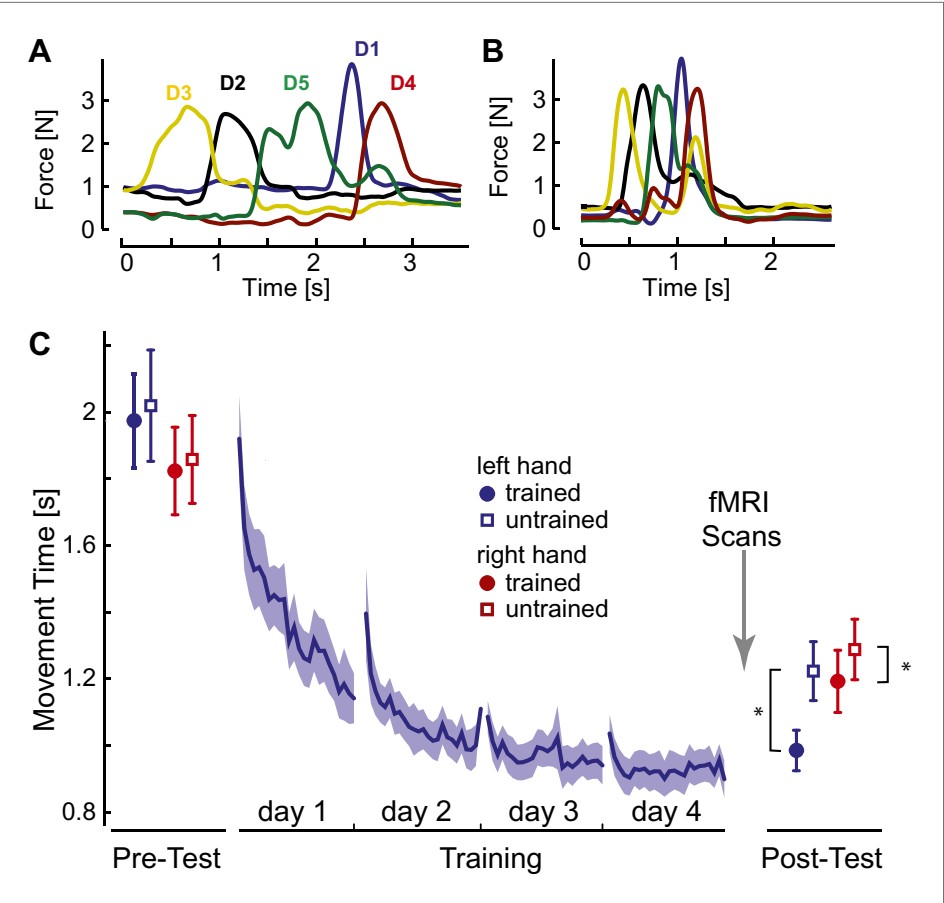

**Figure 2**. Behavioral consequences of sequence learning. (**A**) Before learning, the finger sequence was executed in a slow, deliberate fashion. The force traces of one exemplary trial (sequence: 3-2-5-1-4) are shown. (**B**) After training, the same sequence is produced much faster, with individual finger presses overlapping. (**C**) Group-average MT for the left hand (blue line) reduces during training. In the pre- and post-test, the left (blue) and right (red) hand was tested on trained (filled circle) and untrained (empty square) sequences. The results show general learning (reduction in MT for all conditions), and sequence- and limb-specific learning (stronger reduction for the trained sequences on the left hand). Stars indicate significant differences after correction for pre-test differences.

We also tested the mirror-symmetric versions of the trained sequences on the untrained, right hand to determine the degree of intermanual transfer. The right hand showed a pre- to post-test drop of 570 ms (±80 ms) for untrained sequences, $t(15) = −7.13$, $p<0.001$, but also a sequence-specific advantage of 95 ms (±31.45 ms), again significant after correcting for pre-test differences, $t(14) = 2.802$, $p=0.014$. Thus, participants acquired a sequence-specific skill representation that also enhanced performance of the untrained hand.

## Average activation is reduced for trained relative to untrained sequences

Participants were scanned twice between the end of training and the post-test (*Figure 2C*). During one session participants performed the four trained sequences, and during the other four untrained sequences, in both cases with their left hand. The order of scans was counterbalanced between participants. Within each imaging run, the sequences were executed in pseudo-random order, with each trial lasting 13.5 s (*Figure 1D*). This design allowed us to measure repeatedly the activity pattern for each individual sequence. Using traditional univariate analysis, we first determined the regions that showed sequence-related activation increases compared to rest, average over the four sequences. As expected, we found significant activation (*Figure 3A*) in contralateral primary motor (M1) and sensory cortices (S1). Additional bilateral activation was found in secondary motor areas, including dorsal and ventral premotor cortex (PMd, PMv), supplementary motor areas (SMA/pre-SMA), in regions on the

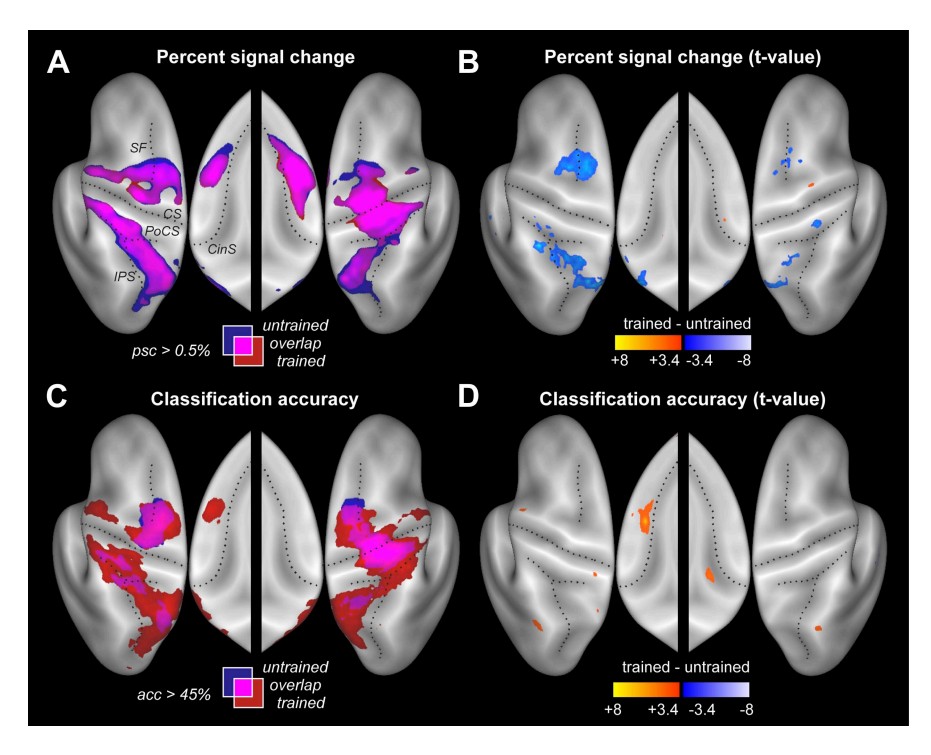

**Figure 3**. Neural differences between trained and untrained sequences. (**A**) Percent signal change compared to rest displayed on an inflated lateral surface of the left and right hemisphere, and on the superior aspects of the medial surfaces (insets). Maps show group-averaged data thresholded at 0.5%, superimposed for trained (red) and untrained (blue) sequences. Purple areas are activated for both. **Figure 3—figure supplement 1** shows the maps separately. CS, central sulcus; PoCS, post central sulcus; SF, Superior frontal sulcus; CinS, cingulate sulcus; IPS, intraparietal sulcus. (**B**) Direct statistical contrast (t-values) of trained sequences against untrained sequences, thresholded at $t(15) > 3.39$, $p<0.002$. Red areas indicate higher values for trained, blue areas higher values for untrained sequences. (**C**) Group-averaged classification accuracy maps (threshold at 45% correct, $Z = 2.57$, $p<0.005$) indicate regions in which the four sequences are associated with significantly different local patterns of activity. (**D**) Direct statistical contrast for classification accuracy, displayed as in **B**.

The following figure supplements are available for figure 3:

**Figure supplement 1**. Separate activity and accuracy maps for trained and untrained sequences.

medial bank of the intraparietal sulcus (IPS), and in the occipital-parietal junction (OPJ) (**Culham and Valyear, 2006**). All activations were highly significant after correction for multiple tests. The pronounced ipsilateral activation during non-dominant hand movements is consistent with the suggested special role of the left hemisphere in complex movements (**Verstynen et al., 2005**).

We then compared the average activity for trained and untrained sequences. Our task instructions regarding speed were designed such that the error rate for the two sequence types during scanning was exactly matched (see methods), therefore equating difficulty and attentional demands. As a consequence, trained sequences were produced at a slightly faster pace. In addition, participants executed the trained sequences with higher peak forces than the untrained sequences (see **Table 1**).

Despite these faster and harder presses, trained sequences were not associated with more activity. Rather, a direct contrast only revealed areas with less activity for trained vs untrained sequences (**Figure 3B** and **Table 2**). Lower activity for trained sequences was found in both hemispheres in PMd and in areas along the IPS. This may indicate that these regions were less involved in the production of trained sequences, or that they were equally involved, but encoded trained sequences more efficiently. Interestingly, the signal decreases (averaged over all fronto-parietal motor regions) were more pronounced in the left than in the right hemisphere, $t(15) = 2.689$, $p=0.017$. No cortical or subcortical area (the cerebellum

**Table 1.** Behavioral performance variables during the fMRI sessions for the trained and untrained sequences. A paired *t*-test for the difference between sessions is reported

|  | Trained | Untrained | t(15) | p |
|---|---|---|---|---|
| MT (ms) | 1209 (297) | 1341 (286) | −3.37 | 0.004 |
| Force (N) | 4.44 (0.63) | 4.0 (0.49) | 3.09 | 0.007 |
| Error rate (%) | 12.26 (6.75) | 12.22 (4.57) | 0.02 | 0.986 |
| ACC (%) | 58.98 (17.86) | 60.55 (11.11) | −0.35 | 0.732 |

MT is the total movement time in ms. The Force is the maximal force produced for each finger, averaged across the fingers. Error rate (%) indicates the percentage of trials containing an incorrect finger press. Accuracy is the classification obtained when distinguishing the four trained or untrained sequences based on MT, force, and error rate.

was not covered) showed significantly more activity during the production of the trained compared to untrained sequences.

## Multiple motor and premotor regions show sequence-specific activation patterns

We predicted that in regions involved in the production of motor sequences, individual movement sequences would be associated with distinct patterns of cortical activity. We tested this idea using a surface-based searchlight (see 'Materials and methods') in which we sequentially selected small areas of neocortex (circular regions of ~10 mm diameter) and examined classification accuracy in each. A linear classifier determined whether the spatial fMRI activation patterns in each area reliably differed between the four trained sequences—and between the four untrained sequences. We then assigned the cross-validated classification accuracy to the center of the region, thereby building a map of sequence representation across the whole cortical surface.

The resultant classification map shows a widespread above-chance accuracy, even for untrained sequences (*Figure 3C*, blue/purple, *Figure 3—figure supplement 1*). Classification was best in right M1, bilaterally in PMd, and on the medial bank of the IPS, where group-averaged accuracies reached ~55% (chance performance was 25%). We also found significant classification in PMv, SMA, and pre-SMA, each of which was highly significant after correcting for multiple tests, even though the classification accuracies were somewhat lower in these regions.

Does above-chance classification accuracy imply that different sequences were represented as distinct spatial activation patterns as hypothesized (*Figure 1B,C*)? One may argue that a region that contains neuronal populations that encode single finger movements (*Figure 1A*) may have been activated

**Table 2.** Cortical areas showing significantly less average activation for trained than untrained sequences. The opposite contrast did not result in any significant areas

| Area | Area (cm²) | P_cluster | Peak t_(15) | MNI X | Y | Z |
|---|---|---|---|---|---|---|
| Left hemisphere | | | | | | |
| PMd | 7.72 | <0.002 | 6.02 | −27 | −1 | 48 |
| OPJ | 1.43 | <0.002 | 6.74 | −10 | −67 | 59 |
| IPS | 0.16 | 0.020 | 7.55 | −42 | −43 | 40 |
| Right hemisphere | | | | | | |
| PMd | 0.14 | 0.020 | 4.25 | 37 | 16 | 51 |
| IPS | 0.13 | 0.028 | 5.28 | 19 | −49 | 61 |
| IPS | 0.27 | <0.002 | 5.48 | 29 | −52 | 49 |

Table shows the result of a surface-based random effects analysis (N = 16). The uncorrected threshold was p<0.002, t(15) > 3.39, and P_cluster is the p-value corrected for multiple comparisons over the whole cortical surface using the area of the cluster (*Worsley et al., 1996*). The coordinates reflect the location of the cluster's peak in MNI space.

with slightly different time-courses for the different sequences, which may have been picked up by the classifier. This explanation, however, is unlikely for two reasons: first, the spatial distribution of classification accuracy for sequences differs substantially from that for single finger movements (*Diedrichsen et al., 2012*), especially in the ipsilateral, left hemisphere (*Figure 4A*). Here, single finger movements are represented close to the central sulcus, whereas sequences are represented more prominently in premotor and parietal areas. Secondly, to test the idea of different temporal activation profiles directly, we defined six bilateral regions of interest (ROIs; see *Figure 5A* and 'Materials and methods'). Within each region, we identified the main components of the temporal response (see 'Materials and methods'). The first component yielded the canonical activation profile, while the additional components described temporal variations around this mean time course (*Figure 4B*). Classification accuracy was highest when using the weight of the first temporal component for each voxel as input to the classifier. When we added the voxel-by-voxel weights for any further temporal component, classification accuracy reduced markedly, indicating that these components did not carry any additional information (*Figure 4C*).

The temporal analysis also allowed us to determine how activation and classification accuracy evolved over the course of a trial (*Figure 4D,E*). While the experimental was not designed to distinguish between instruction- and execution related activity, two insights can be gathered from the analysis. First, classification accuracy for S1 was already above chance for the second TR, which was acquired 2.7–5.4 s after the onset of the number cue, $t(15) = 3.335$, p=0.002. All other regions showed significant classification accuracy at the third TR, all $t(15) > 5.34$, p<0.0001. Thus, it is likely that the informative patterns were caused at least partly by instruction-related activity. Secondly, even areas such as OPJ and PMd exhibited sustained classification accuracy over the whole trial. Thus, these areas did not only represent the string of numbers presented on the screen; rather they exhibited a sustained representation of the sequence during the execution phase, during which the number cue was not visible anymore.

Importantly, our hypothesis predicted that each sequence would be associated with a unique spatial activation pattern. To investigate this directly, we employed a newly developed method that consists of a set of classifiers, each using a different number of informative spatial pattern components (*Diedrichsen et al., 2013*). To illustrate this method, assume that each sequence activated a unique set of voxels. To describe the differences between these four patterns, one would need three linear components (or contrasts), each of which would capture an equivalent amount of variance. Thus, the patterns would be distributed evenly in the space spanned by the three pattern components (*Figure 5A*, red dots). Consequently, classification accuracy should be highest when taking all three components into account. In contrast, if sequences could be distinguished because they differed on a single variable (e.g., difficulty), then they should be associated with a single activity pattern that is scaled in intensity by different amounts. The patterns would therefore mostly differ along a single pattern component (*Figure 5A*, blue dots). In this case, classification accuracy should be highest when using only the most informative spatial component (one-dimensional simulation, *Figure 5B,C*). While we have shown that this is indeed the case when the same movement is executed at different force levels (*Diedrichsen et al., 2013*), we found here that all sequence representations showed the highest accuracy when using all three available linear components (*Figure 5B,C*, red line). This was the case in all ROIs and for both untrained sequences (all $t(15) > 2.496$, p<0.025) and trained sequence (all $t(15) > 5.261$, p<0.001). Thus, we can conclude that sequences are encoded in unique spatial activation patterns, rather than in differently scaled versions of a single pattern.

## Sequence-specific activation patterns become more distinct with learning

Our central prediction was that the distinctiveness of cortical activation patterns, measured by classification accuracy, should increase with training. This is indeed what we found. We first tested this idea globally, averaging over all fronto-parietal cortical regions. For trained sequences, classification accuracy (*Figure 3C*, red/purple, *Figure 3—figure supplement 1*) reached 60%—and was on average significantly higher for trained (38.4%, ±1.39%) than for untrained sequences (34.2%, ±1.76%), $t(15) = 2.203$, p=0.022. Furthermore, the cortical surface area which encoded the sequences better than chance (within subject threshold: acc > 37.5%, Z > 1.64), increased from 35.12 cm² (±5.38 cm²) to 46.93 cm² (±4.65 cm²), $t(15) = 1.963$, p=0.034. Finally, we also conducted a map-wise comparison between trained and untrained sequences for the classifier using all three spatial components (*Figure 3D*). After correcting for multiple comparisons over the whole cortical surface, only the increase in left SMA/pre-SMA was significant (uncorrected threshold p=0.002, $t(15) > 3.39$; p-corrected < 0.012; clustersize = 0.19 cm²). In this area, the accuracy increased from 36% to 46% (*Figure 5C*).

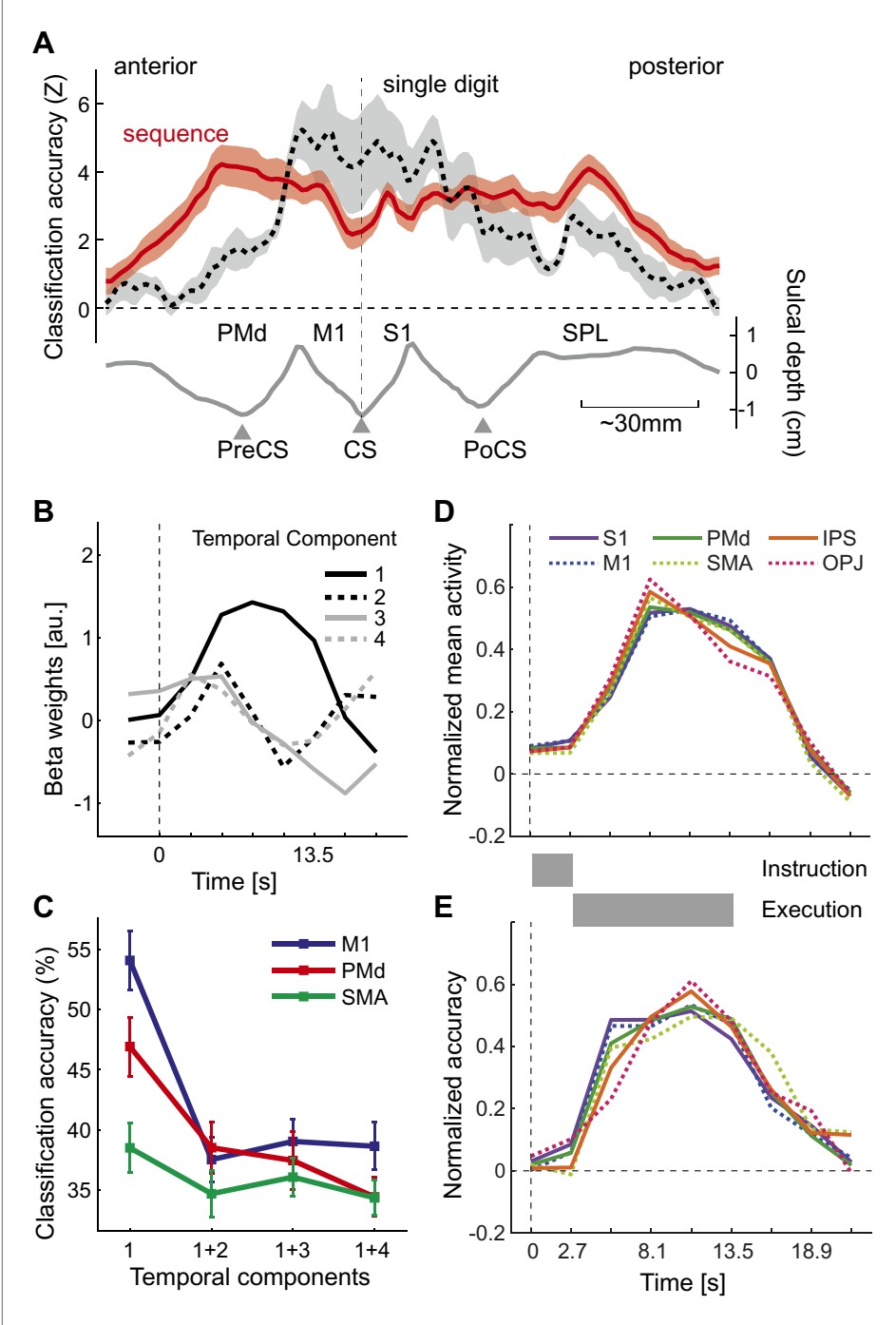

**Figure 4**. Temporal aspects of sequence representations. (**A**) Spatial distribution of classification accuracy in the left hemisphere for trained left-hand sequences (red) differs from left-hand single digit movements (dashed, *Diedrichsen et al., 2012*). Shown is a cross-section through the surface map, from the rostral end of dorsal premotor cortex to the posterior superior parietal cortex. The lower curve indicates the average sulcal depth, showing the location of the central sulcus (CS), postcentral sulcus (PoCS) and precentral sulcus (PreCS). (**B**) The four most informative temporal components of the BOLD response, shown exemplary for right M1. (**C**) Classification accuracy in three ROIs, using either only the first temporal component, or using the first and one additional temporal component. Adding further temporal components does not improve classification accuracy. (**D**) Normalized time course of average activation across the time course of a trial for six bilateral ROIs. Length of instruction and execution phase are indicated as gray bars. (**E**) Normalized classification accuracy over the time course of the trial.

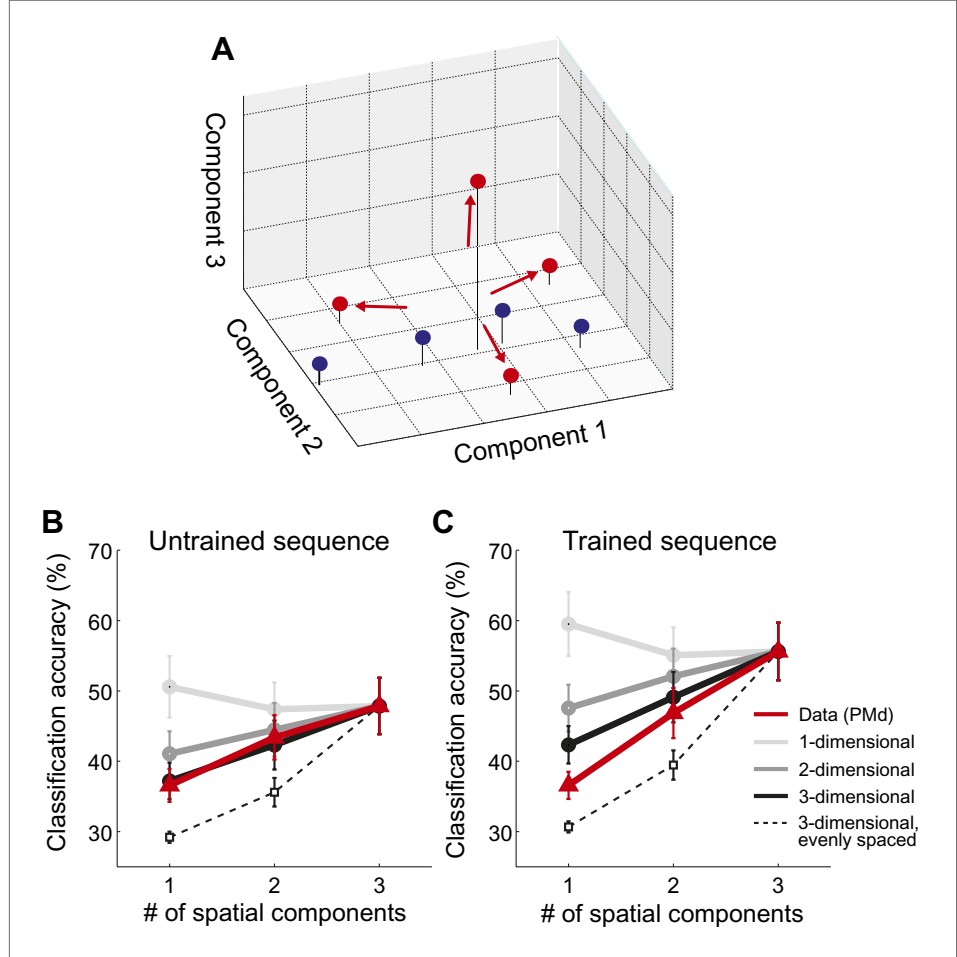

**Figure 5**. Spatial dimensionality analysis. (**A**) Hypothetical distribution of activity patterns in the space of spatial pattern components. One set of activity patterns (blue dots) may differ mostly in the intensity of a common pattern component, and should be distinguished relatively well along this single component. Another set (red dots) may consist of four unique activity patterns and should therefore differ also along the second and third pattern component. (**B** and **C**) Classification accuracy in right PMd, using 1, 2, or 3 pattern components (red line) for untrained (**B**) and trained (**C**) sequences. Expected accuracies are derived from simulations using patterns randomly spaced in 1–3 dimensional space (light-dark gray) and evenly spaced in three dimensions (dashed line). Each simulation matched the data for the accuracy of the one-dimensional classifier (***Diedrichsen et al., 2013***).

The dimensionality analysis (***Figure 5***; ***Diedrichsen et al., 2013***) allows further insight into the representational changes that occur with training. Specifically, by looking at how the sequence-specific patterns are distributed in the space of pattern components, we can determine whether sequences activated separate or overlapping sets of voxels. Our data indicates that untrained sequences were associated with unique, but basically random activity patterns. This means that any two activation patterns exhibit some non-overlapping features, but also share a specific amount of shared activity. The predicted accuracy curve when classifying four simulated random patterns using 1–3 spatial components is shown in ***Figure 5B,C*** (black line). In each simulation we matched the accuracy for the three-dimensional classifier to the empirically observed accuracy. The resulting prediction for the 1- and two-dimensional classifier characterized the data for untrained sequences quite well (***Figure 5B***). Only in the IPS was the observed classification accuracy for the one-dimensional classifier lower than predicted, $t(15) = 2.247$, $p=0.040$; for all other ROIs, the prediction and data did not differ significantly (all $t(15) < 1.206$, $p>0.247$).

For the trained sequences, we reasoned that practice should lead to the development of dedicated neuronal populations for trained sequence transitions. This development would cause each sequence

to activate a unique subset of the network, that is reduce the overlap between patterns. In the extreme case, in which each sequence activates an exclusive set of voxels, the patterns would become evenly distributed in the space of possible patterns (red dots, *Figure 5A*). The accuracy curves based on such an arrangement is shown as the dashed line in *Figure 5C* (for details, see *Diedrichsen et al., 2013*). While the observed classification accuracy curves for trained sequences did not fully reach this extreme prediction, the activity patterns were more evenly distributed than expected for spatially random patterns: The classification accuracy for the one-dimensional classifier deviated significantly from the prediction based on random patterns for S1, M1, PMd, SMA/pre-SMA, and IPS, all $t(15) > 2.429$, p<0.028.

We then compared the accuracy curves for trained and untrained sequences directly (*Figure 6*). For a classifier relying only on a single dimension, no significant differences were found. However, when considering also the second and third component, classification accuracy increased more for the trained sequence. When correcting for the multiple tests over the six ROIs, the effect was significant in the SMA/pre-SMA, S1, and in IPS ($F(2,30) > 5.64$, p<0.0083), but we also found a similar trend in the remaining regions ($F(2,30) > 3.32$, p<0.05, uncorrected). The effect was bilateral–in none of the tested regions was there a significant interaction of this effect with the hemisphere (all $F(2,30) < 2.072$, p>0.144).

Could these accuracy differences be an artifact of performance differences during the scan? While trained and untrained sequences were executed at slightly different speeds and average forces (*Table 1*), the individual differences in overall classification accuracy did not correlate with the difference in MT (p=0.75), force (p=0.51), or error rate (p=0.21). Furthermore, we tested whether the increase in discriminability of cortical activation patterns could have been caused by the reduced behavioral variability that normally accompanies learning. In considering this idea, it is important to keep in mind that classification accuracy is determined both by within-sequence variability and by between-sequences differences. To evaluate both together, we tested how well a linear classifier could discriminate between each set of four sequences based on MT, average force, or error rate, either considered in isolation or in any combination as separate features. None of the seven combinations showed a significant difference between trained and untrained sequences (all $t(15) < 1.15$, p>0.269; see *Table 1*). Furthermore, when including the differences in classification accuracy based on all behavioral variables as a covariate, the accuracy difference based on fMRI patterns remained significant, $t(15) = 2.56$, p=0.01. Thus, the higher classification accuracy for trained sequences was not a simply a consequence of more stable behavioral performance.

In summary, our study shows for the first time that the sequence-specific component of the activity patterns increases in strength with training. While a recent report (*Huang et al., 2013*) showed increases in split-half correlation of the activity pattern in primary motor cortex for a trained compared to an untrained finger sequence, these authors only used a single trained sequence, and therefore could not distinguish between the component of the activity pattern that is common to any possible trained sequence, and the pattern component that is specific to a single sequence. This, however, is an important distinction: when decomposing our activity patterns (*Diedrichsen et al., 2011*), we found that 99% of the voxel-by-voxel variance was explained by a pattern component that is common to all trained sequences, with less than 1% being attributable to the sequence-specific component (*Figure 7*). In most areas, the sequence-specific component was larger for trained sequences, and therefore followed the pattern observed in the classification accuracy. These differences, however, failed to become significant. In contrast, the common component was smaller for trained than untrained sequences (significant for PMd, $t(15) = 2.469$, p=0.026, and IPS and OPJ, $t(15) = 3.573$, p<0.003) and therefore followed the pattern found for the average activity (*Figure 6*).

Here we did not find larger split-half correlations for the patterns associated with trained compared to untrained sequences. We believe that this discrepancy is caused by the fact that a split-half correlation of a single activation pattern mostly depends on the much stronger common component, which was found to be smaller for trained than untrained sequences in our study, but slightly higher in the study by *Huang et al., (2013)*. This underlines the importance of using experimental methods that allow a separation between the unspecific (common) and information-carrying aspects of neural activity patterns.

## Discussion

Together, our results show that different sequences of the same five finger presses are associated with overlapping but discriminable patterns of activity, and that these differences are more pronounced

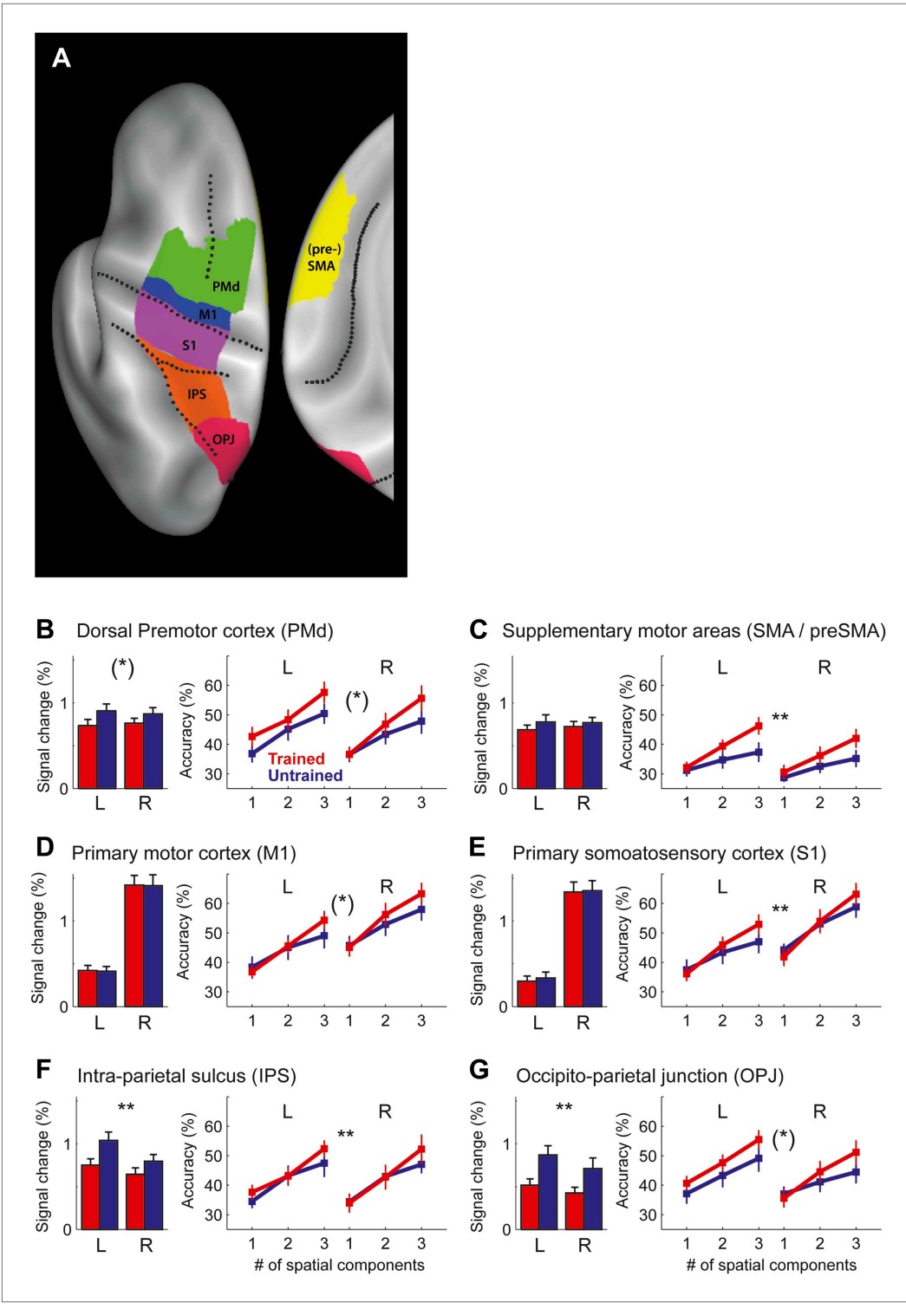

**Figure 6.** Region of interest (ROI) analysis. (**A**) ROI definition on the average cortical surface (shown in a dorsal and medial view), based on anatomical criteria (see 'Materials and methods'; **Fischl et al., 2008**). Regions of the right hemisphere are defined in a symmetric fashion. (**B–G**) Left panel shows the average percent BOLD signal change for the left and right hemisphere. Stars indicate a significant difference between trained (red) and untrained (blue) sequences (** corresponds to $p < 0.05/6$, or $p < 0.05$ corrected for multiple comparisons, [*] indicates $p < 0.05$, uncorrected). Right panel shows classification accuracy for the classifiers using 1–3 of the most informative spatial dimensions (**Diedrichsen et al., 2013**). Stars indicate a significant interaction effect of number of spatial components and sequence type (trained/untrained), p-levels as above.

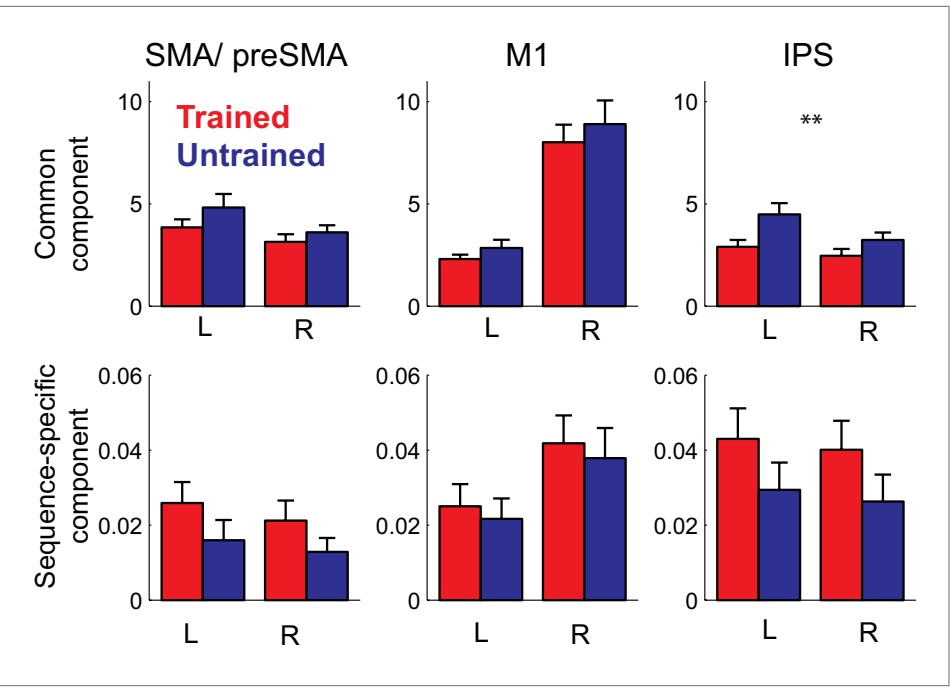

**Figure 7**. Pattern decomposition analysis for 3 of the 6 anatomically defined ROIs. Activity patterns are decomposed into a component that is common to all four trained or untrained patterns, a sequence specific component, and a noise component (***Diedrichsen et al., 2011***). Plotted is the estimate of the voxel-by-voxel variance for each component, relative to the variance explained by noise. The variance explained by the common pattern outstrips the variance associated with the sequence specific component by a factor of 100 or more. While the strength of the common component follows the mean activation (***Figure 5***), the sequence-specific component shows the same advantage for the trained compared to the untrained sequences evident in the classification accuracy.

when the specific sequences are trained for multiple days. The changes likely reflect the formation of specialized neuronal circuits that enable fast and accurate sequential movements.

We hypothesize that the differential activation patterns reflect, at least partly, populations of neurons with preferential tuning for the sequential context of movements. Neurons in M1 (***Matsuzaka et al., 2007***) and SMA and pre-SMA (***Tanji and Shima, 1994***) have been reported to be preferentially active during performance of specific movement transitions. The latter study also found cells that discharged at the beginning of specific longer sequence, but not for other orderings of the same movements. Regions that contain either type of neuron will therefore activate slightly different neuronal populations depending on movement order.

To be visible in fMRI, the patterns of presynaptic input to these neuronal populations must differ on a relatively coarse spatial scale. Although MVPA is able to decode information that relies on fine-grained spatial details of activity patterns (***Swisher et al., 2010***; ***Freeman et al., 2011***), it was not clear a priori that sequential tuning would be clustered enough to reveal such representations at standard fMRI resolution. However, we found a set of areas, highly replicable across subjects, with activity patterns that distinguished reliably between sequences.

In these areas we could significantly distinguish even between the four untrained sequences. This finding may be related to the substantial generalization of learning to the untrained sequences. For example, classification may have relied on populations of neurons that encode for the sequential transitions between pairs of fingers, many of which were shared between trained and untrained sequences. More likely, however, the unique activity patterns for untrained sequences relied on preexisting neuronal coding of sequential context: whether typing on a keyboard or playing an instrument, humans engage in many sequential motor behaviors, and such general representations would discriminate between trained and untrained sequences alike.

We found sequential representations in motor areas in which single-cell neurophysiology has revealed such encoding before (i.e., M1 and SMA; ***Matsuzaka et al., 2007***; ***Tanji and Shima, 1994***),

but also in a set of parietal and lateral premotor areas. While other imaging studies have reported increased activity in some of these regions with increased sequential processing demands (*Farooqui et al., 2012*; *Heim et al., 2012*), our study is the first to show that individual voxels are tuned for different sequence of movements. Together with other neurophysiological recording study (*Fogassi et al., 2005*), our results argue that sequential encoding is not unique to the SMA, but is rather widespread throughout the cortical motor hierarchy.

The dimensionality analysis (*Figure 5*) indicated that each of these areas does not simply discriminate between sequences based on one single variable (difficulty, force, etc), but that each sequence is associated with a unique activity pattern. The exact response characteristics of the underlying neuronal populations remain unclear, however. Some of these regions could represent the sequences phonologically as string of number during subvocal rehearsal (*Hartwigsen et al., 2013*), abstractly in a spatial reference, or motorically in an intrinsic references frame (*Keele et al., 1995*). Furthermore, neural populations could code for the order of finger presses or their relative timing (*Kornysheva et al., 2013*), and representations could be effector-specific or independent of hand that is used to execute the movement (*Gallivan et al., 2013*). Our experimental design also did not allow us to determine conclusively to what degree sequential decoding relied only on activity related to the execution, or to what degree these patterns were also present during the instruction phase. Further dedicated experiments are needed to identify the exact nature of the sequential representations in identified areas.

Our main hypothesis, however, was that training promotes the formation of specialized neural representations and hence leads to more distinguishable activity patterns. Indeed, averaged across all involved areas, we were better able to classify between trained than between untrained sequences. Furthermore, we could show that these training-related changes became most evident when looking at multiple spatial components of the activity patterns. This finding implies two things. First, it shows that the changes in classification accuracy were not due to larger between-sequence differences, or lower within-sequence variability, of a single behavioral factor—otherwise the difference would have been apparent using a classifier that uses only the most informative spatial pattern component. Secondly, it shows that training leads to the formation of a unique spatial activity pattern for each sequence and therefore a more even spacing of the activity patterns in the space of possible patterns. While it is still possible that this observation reflects changes in a combination of multiple other behavioral variables, none of the possible combinations of the behavioral measures was able to explain the observed accuracy difference between trained and untrained sequences. This lends some credibility to the idea that learning caused by the formation of specialized neuronal circuits that encodes different finger transitions or whole sequences.

The largest (and in a whole-brain comparison most reliable) difference in classification accuracies was observed in the left SMA and pre-SMA. This likely reflects neuronal circuits specialized for longer sequence segments, a representation that would be highly specific to the trained sequences. Other areas, such as M1, also showed good discrimination for the untrained sequences, possibly reflecting representations of shorter sequence elements. In the framework of hierarchical sequence representations (*Kiebel et al., 2009*), sequence-specific units in SMA may trigger the shorter sequence elements in M1.

A number of previous studies have highlighted the SMA and pre-SMA as critical structures for skillful production of motor sequences. Disruption of the mesial frontal cortex caused increased errors during performance of complex sequential finger movements (*Gerloff et al., 1997*), while leaving performance of simple finger sequences unaltered. A second study showed slowing in the transition between different sequences after pre-SMA stimulation (*Kennerley et al., 2004*). In non-human primates, muscimol injections into SMA and pre-SMA lead to errors in memory-guided performance of arm movement sequences (*Shima and Tanji, 1998*).

Although we trained and tested the left hand, learning-related changes were either bilateral, especially in regions that appear to code for actions with both hand (*Gallivan et al., 2013*), or in some cases even more pronounced in the ipsilateral, left hemisphere. The latter finding is consistent with specialization of the left hemisphere for complex movements, executed with either the dominant or non-dominant hand (*Verstynen et al., 2005*). Indeed, our behavioral measures indicate inter-manual transfer of the sequence-specific knowledge to the untrained right hand.

The increases in discriminability of cortical activation patterns contrasted drastically with changes in average activity. Generally, we observed less activity for trained than untrained sequences, despite the fact that the two conditions were matched for error rate (and hence difficulty) and the trained sequences were executed slightly faster and with higher peak forces. In areas that showed the largest increases in classification accuracy, no changes in average activation were found.

Average activity changes during learning are likely complicated by the overlap of two competing tendencies. First, training likely leads to increased recruitment of neuronal populations, as has been shown with expansion of the cortical hand area following skill training (*Nudo et al., 1996*). Involving larger populations of neurons in motor planning and execution may reduce neuronal noise and lead to less variable performance. Second, the recruited neurons likely become more specialized for specific trained behaviors. Therefore, fewer non-specialized units must be recruited, decreasing the activity for the majority of neurons in a region (*Poldrack, 2000*). The overlap of increased neuronal recruitment and increased efficiency through specialization (*Steele and Penhune, 2010*) may explain previously inconsistent results, which suggest that activity in sequence-related regions increases (*Grafton et al.,1995*; *Karni et al., 1995*; *Floyer-Lea and Matthews, 2005*; *Hazeltine et al., 1997*), decreases (*Wu et al., 2004*; *Poldrack et al., 2005*), or changes non-linearly over the course of learning (*Xiong et al., 2009*; *Ma et al., 2010*).

In contrast, the representational analysis employed here directly reflects the increased specialization of cortical circuits for trained behaviors. Although average activity decreased in most areas, we found increased bilateral encoding of the trained sequences. This finding has important implications for models of skill development. Examining average activity alone, one might be tempted to conclude that many secondary motor areas play a reduced role in the production of highly trained skills, which may be encoded in a few execution-related areas such as the primary motor cortex (*Penhune and Steele, 2012*). In contrast, our results indicate that skill learning generates increasingly specialized representations, which are distributed widely across both primary and secondary cortical motor areas.

## Materials and methods

### Participants

Eight female and eight male healthy, right-handed volunteers (22.4 years, SD = 2.6), participated in the experiment. The UCL Ethics Committee approved all study procedures.

### Apparatus

We used an fMRI-compatible response box, equipped with five piano-like keys, each incorporating a sensor (FSG-15N1A; Sensing & Control Honeywell Inc., Morristown, NJ) that continuously measured isometric forces during sequence production. The force measurements were transmitted to a control computer outside the scanning room through a set of filtered cables to prevent RF-signal leakage into the MR-environment.

During both training and scanning participants lay supine on a (mock-) scanner bed, with the keyboard firmly placed on their lap at a 45° angle. Participants received visual instructions and feedback though a back-projection screen, viewed though a mirror. A central asterisk served as fixation cross throughout training and scanning.

### General procedure

Each trial consisted of either five (training) or three (scanning) repetitions of the same sequence. At the beginning of each trial, the sequence was announced by five centrally presented numbers for 2.7 s (*Figure 1D*). Each number referred to a digit (one for thumb, five for little finger). During the announcement, participants were instructed to memorize the sequence.

The digit string was then replaced by the fixation cross. Simultaneously, five white asterisks were presented above the box. This display served as the starting signal to produce the sequence as fast as possible. A keypress was recognized when the force of a finger exceeded a threshold of 2.5 N, while the other fingers were below 2.2 N. If the correct finger was pressed, the corresponding asterisk in the sequence turned green. If a participant pressed the wrong finger, the asterisk turned red instead. Participants were instructed to complete the sequences even if they made an error and to keep their fingers placed on the respective keys at all times.

After completion of a single sequence (five presses), the central fixation-cross changed color. 'Green' indicated correct sequence production (one point), 'red' that one or more errors occurred (-one point), and 'blue' that the sequence was produced 20% slower than the median MT in the previous run (zero points). To motivate participants during the training, we also presented three green stars if the sequence was produced 20% faster than the median MT (three points). After the end of this short feedback phase (800 ms), all asterisks turned white again to signal the start of the execution. After the required number of sequences was performed, the trial ended and the next sequence was announced. During scanning only error feedback and feedback for slow executions were provided, and the frequency of each feedback type was matched between trained and untrained sequences.

Based on a pilot experiment with N = 5 independent participants, we selected a total of 12 different finger sequences with approximately the same difficulty. Each sequence contained each of the five fingers once and differed only in their order. None of the sequences contained an ascending or descending sub-sequence of more than three neighboring fingers. For each participant, these 12 sequences were randomly divided into three sets of four sequences: one to be trained, one to be used as untrained control sequences for the pre- and post-test, and one as the untrained control sequences for the scan.

## Behavioral testing and training

The experiment started with a short familiarization phase, which was followed by a pre-test. Here we measured how well participants performed eight sequences with the left and right hands. For right-hand performance, the sequences were performed in mirror-symmetric fashion, for example, pressing the thumb for the number 1, etc. The pre-test contained 36 trials, with each sequence repeated two times per hand (10 executions total). We counterbalanced sequence order by presenting the same sequences in the reverse order in the second half of the pre-test. Participants were then trained to perform the sequences in the training set with the non-dominant left hand. On each of the four separate training sessions, they performed 24 runs (96 trials, and 480 sequence executions). The sessions were usually separated by 24 hr, with a few exceptions in which there was a 48-hr gap. A day after the second scanning session (see below) we conducted a post-test that had exactly the same format as the pre-test.

After each run, feedback about error rate, average MT and points was presented. We instructed participants to decrease their MT if they had an error-rate of less than 20% and to focus on accuracy if the error rate was larger than 20%. This speed-accuracy instruction served to keep error-rates stable across the experiment.

To assess sequence-specific learning, we tested whether trained sequences were performed faster than untrained sequences at post-test. To correct for possible pre-test differences between sequences, we calculated a regression between the pre-test difference (x-variable) and the post-test differences (y-variable) and tested whether the intercept was significantly different from zero.

## Scanning procedure

After 4 days of sequence training, participants underwent two sessions of fMRI scanning on separate days. During one session, participants performed the four trained sequences, and during the other, four novel sequences that were not tested in pre- or post-test. The order of these sessions was counterbalanced between participants. Each imaging session comprised 8 runs of 16 randomly ordered trials, 4 per sequence. Each trial consisted of an announcement phase of 2.7 s and three sequence executions (*Figure 1D*). To synchronize the paradigm to image acquisition, participants had a maximum of 2.8 s to complete each sequence. There were also four, randomly interspersed rest phases (13.5 s each) in each run.

## Scan acquisition

Imaging data were acquired on a 3T Siemens Trio MRI scanner using a 32-channel head coil. For each participant we also obtained an anatomical image (3D MPRAGE sequence, 1 mm isotropic). Functional data were acquired using a two-dimensional echo-planar sequence (TR = 2.72). Each functional scanning session consisted of 8 runs of 110 vol each. The first three images of each sequence were excluded from the analysis. We acquired 32 slices with 2.15-mm thickness in an interleaved sequence (0.15 mm gap, 2.3 × 2.3 mm$^2$ in-plane resolution) in an axial orientation. This arrangement covered the dorsal part of both cerebral hemispheres, but not the inferior temporal lobe or the cerebellum. To correct for distortions arising from field inhomogeneities, we also acquired a B$_0$ field-map with the same slice prescription as the functional data (*Hutton et al., 2002*).

## Imaging data analysis

The imaging data were analyzed using SPM8 (http://www.fil.ion.ucl.ac.uk/spm/), and custom written MATLAB routines (The MathWorks, Inc., Natick, MA). Preprocessing consisted of correction for field inhomogeneities (*Hutton et al., 2002*), motion realignment, high-pass filtering (cut-off frequency of 1/128 s), and co-registration between functional and individual anatomical data.

To measure the signal changes for each voxel during sequence performance, we modeled the unsmoothed data using a general linear model. We defined a unique regressor for each sequence per run. These regressors were boxcar-functions (length 13.5 s), convolved with a standard hemodynamic response function approximated by the sum of two Gamma-functions (spm_hrf.m in SPM8). The regression-coefficients were then estimated using robust linear regression (*Diedrichsen and Shadmehr, 2005*), correcting for movement artifacts by down-weighting noisy images. The regression coefficients indicated the size of the activity change for each specific sequence and were used as the input to both the traditional univariate analysis and MVPA.

## Multi-voxel pattern analysis

To determine whether a specific area of cortex showed reliably different patterns of activity for the four tested sequences, we sequentially selected 160 voxels contained within a spherical patch of cortex (see surface-based searchlight below) and then submitted these to a linear discriminant analysis (LDA; *Duda et al., 2001*). The input data ($x_i$) consisted of 4 (sequences) × 8 (runs) activation estimates for the p=160 neighboring voxels. Using the data from seven runs, we calculated the mean activation vector for each sequence, and the average *PxP* within-class covariance matrix Σ, which was regularized by adding 1% of the diagonal mean to all diagonal elements. The activation vectors from the remaining eighth run were then classified by assigning them to the class with the highest likelihood *p*(x) (for details, see *Diedrichsen et al., 2011*). By retraining and cross-validating the classifier with all possible training and test sets, we obtained an average classification accuracy. If the neural activation patterns did not differ systematically between sequences, accuracy should be 25% (guessing rate). Systematically higher classification rates indicate that a region showed differential activation patterns for the four sequences, and the size of the classification accuracy served as a measure of the strength of the sequence representation in that region. For between-subject analysis, we z-transformed the classification accuracies, using a normal approximation to the binomial distribution.

To determine whether differences in classification accuracy could be caused by lower variability in the behavioral performance—or by larger behavioral differences between individual sequences, we also performed a LDA on the behavioral data. As for the neural activation, we calculated the average MT, force, and error rate for each sequence and run, z-standardized these, and submitted these in isolation or as separate features (treating each behavioral variable like an individual voxel) to the same cross-validated LDA.

We also applied LDA to different temporal components of the BOLD response. We modified our first-level analysis, such that each trial was modeled with an arbitrary finite impulse response of 24.3 s (9 TRs) length, using 9 boxcar regressors, one for each TR, for each sequence and run. Using single value decomposition we then determined the main temporal components of the response for each anatomical ROI (*Kay et al., 2008*). Instead of the beta-estimate for the canonical hemodynamic response, we submitted the component weights for each voxel and trial to the LDA classifier. The first temporal component reflected the main response and led to the highest classification accuracy in all ROI. We then added each of the other temporal components by using the 160 voxel-weights for first component and the 160 voxel-weights for the additional component as 320 independent features.

We also used the finite-response function model over 9 TRs to determine the time course of activation and information over a single trial (*Figure 4D,E*). For this we either averaged the beta weights for each TR to obtain the average activation time course, or submitted the beta weights for each TR to a cross-validated LDA classifier to obtain a classification accuracy time course. To compare different time courses across regions, we normalized each individual time-series by the L2-norm of their average.

To determine the number of pattern components underlying the difference between patterns, we used a one-, two- and three-dimensional classifier (for details see *Diedrichsen et al., 2013*). In short, to distinguish four unique patterns, at most three linear dimensions (pattern components) are needed. If the patterns only differ in the intensity scaling of a common pattern, then best classification accuracy should be achieved with a classifier that uses only the most informative pattern component of the training data set. If, however, the patterns are evenly distributed in the pattern space, classification accuracy should increase for each additional spatial dimension that the classifier uses. Simulations of classification accuracies under different assumptions are conducted so that the classification accuracies

of the three-dimensional classifier matched those of the data (*Figure 4E*; for details, see *Diedrichsen et al., 2013*).

Finally, we also decomposed the activity patterns of each ROI into a common activity component that is shared between the sequences, a component that is specific to each of the four sequences, a noise component that varies trial-by-trial, and a noise component that is common to all trials within a run. The employed method directly estimates the variability (or strength) of each pattern component across voxels (*Diedrichsen et al., 2011*). Classification accuracy relates tightly to the ratio of the sequence-specific component over the noise component. The decomposition analysis was performed separately for trained and untrained sequences.

## Surface-based searchlight

To detect sequence representations anywhere in the cortex, we used a surface-based searchlight approach (*Oosterhof et al., 2011*). We first reconstructed cortical surfaces for each participant using Freesurfer (*Dale et al., 1999*) and aligned these to a template surface using spherical registration. For each surface node, we selected a surrounding circular region such that p=160 partly touched, or lay between, the pial and grey-white matter surface patches. This resulted in a searchlight radius of 10.4 mm on average. The corresponding classification accuracy was then assigned to the center node. By sequentially selecting each node of the cortical surface, we built up a map of where and how well sequences were represented in the neocortex.

## Regions of interests

We defined six bilaterally defined regions of interest to cover the main anatomical areas that showed encoding for sequences in general (*Figure 6A*). Using probabilistic cytoarchitectonic maps (*Fischl et al., 2008*), only surface nodes that belonged to Brodman area (BA) 4 with maximal probability were included into the M1 ROI. To exclude mouth and leg representations, we further excluded all nodes that had a distance of more then 2.5 cm from the hand knob (*Yousry et al., 1997*). S1 was similarly defined as the hand-related aspect of BA 1,2, and 3. BA 6 was divided into a medial aspect (SMA/pre-SMA) and the lateral aspect superior to the crest of the middle frontal gyrus (PMd). The posterior parietal cortex was divided into an anterior region, including anterior, medial, and ventral IPS, and a posterior region, including the medial and lateral OPJ (*Culham and Valyear, 2006*).

For the analysis of the mean activity, we averaged all voxels within each ROI. For classification and decomposition analysis, we selected within each anatomical ROI and participant the 800 most activated surface nodes, causing each ROI to have a size of approximately 260 voxels. Because the MVPA measures are independent of the mean activity, this selection does not bias the results under the null-hypothesis. Classification analysis within each selected part of the ROI was performed using randomly drawn groups of 160 voxels, repeating this process 5000 times, and averaging the accuracy over all draws. This random-subspace approach increases the reliability of accuracy for ROI-based analyses (*Diedrichsen et al., 2013*).

## Statistical testing and regions of interest

To compare the representation of trained and untrained sequences we employed three levels of inference, each using a random-effect analysis (N = 16). On a whole-system level, we summarized the classification accuracy averaged over all fronto-parietal regions. Because only one test was conducted for each measure, no correction for multiple tests was necessary. Within each ROI we conducted a repeated measurement ANOVA with the factors hemisphere (left vs right), sequence condition (trained vs untrained), and classifier dimensionality (1–3). All *F*-test were corrected for the number of ROIs using Bonferroni-correction. Thirdly, we also tested difference between trained and untrained sequences using a map-wise contrast. The uncorrected threshold was set to $t(15) > 3.39$, p<0.002, and family-wise error was controlled by calculating the critical size of the largest super-threshold voxel that would be expected by chance, using Gaussian Field theory as implemented in the fmristat package (*Worsley et al., 1996*). The same threshold was applied for the test of overall activity differences (*Table 2*).

## Acknowledgements

We thank John Krakauer and Katja Kornysheva for helpful comments.

# Additional information

## Funding

| Funder | Grant reference number | Author |
| --- | --- | --- |
| Wellcome Trust | 094874/Z/10/Z | Jörn Diedrichsen |
| Strategic Award to the Wellcome Trust Centre for Neuroimaging | 091593/Z/10/Z | |
| James S McDonnell Foundation | Understanding Human Cognition Scholar Award 2012 | Jörn Diedrichsen |

The funders had no role in study design, data collection and interpretation, or the decision to submit the work for publication.

## Author contributions

TW, JD, Conception and design, Acquisition of data, Analysis and interpretation of data, Drafting or revising the article

## Ethics

Human subjects: Informed consent was obtained before the beginning of all experiments; all experimental and consent procedures were approved by the UCL research ethics committee, protocol numbers PWD210110BJD (studies of motor control and learning) and 1825/003 (minimum-risk magnetic resonance imaging studies of healthy human cognition).

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
