## [Decision Letter]

Thank you for sending your work entitled “Skill learning strengthens cortical representations of motor sequences” for consideration at *eLife*. Your article has been favorably evaluated by a Senior editor and 3 reviewers, one of whom is a member of our Board of Reviewing Editors.

The Reviewing editor and the other reviewers discussed their comments before we reached this decision, and the Reviewing editor has assembled the following comments to help you prepare a revised submission.

All three reviewers were impressed with the paper, with one describing it as “an elegant […] paper examining how the neural representations of motor sequences change with motor learning”. In particular, the reviewers “applaud the authors for the application of several sophisticated and newly developed analysis techniques to examine the neural basis of sequence skill learning” and conclude that: “This finding is novel and of considerable conceptual relevance, and will inspire future neuroimaging research on motor learning.” While the reviewers were enthusiastic about the manuscript’s potential for publication in *eLife*, they also raised some major concerns that need to be addressed.

1) The authors acquired fMRI data for the learned and non-learned sequences in separate runs, and only apply classification within a fMRI run to test whether one learned sequence could be classified better against other learned sequences or whether one non-learned sequence could be classified better against other non-learned sequences (within-class classification). The authors need to comment on whether their data would also allow classification of learned sequences against non-learned sequences (comparison between classes of sequences as defined by the learning history) and to classify a learned sequence against other non-learned sequences. If such comparisons are not possible using the present dataset, please speculate on what one might expect.

2) The authors could do a better job of weaving some key methodological details throughout their Results section. The authors begin discussing their behavioural results (Figure 2) and basic fMRI findings (in Figure 3) before a reader is made aware of the exact task that was employed. Figure 1 requires more details for a reader to fully understand the details of the task and appreciate the results. The authors should re-write parts of the beginning of the Results section to better explain to the reader the key components of the task.

3) A related problem is that fMRI activity was averaged over the cueing and execution phase. Would the same classification results emerge if classification were applied to the instruction phase and execution phase separately? It would be interesting to present classification results based on the cueing or execution phase separately.

4) Figure 3 should include direct statistical comparisons between trained and untrained data in addition to the indirect comparisons provided in the overlap maps. Indirect comparisons are hard to interpret because “a difference in significance doesn’t necessarily indicate a significant difference”. The manuscript text describes the fact that trained sequences evoke less activity than untrained ones but the direct contrast, so it would be nice to see this in this Figure 3 and with p values (not just %SC values).

5) The authors also find considerable general training effects when subjects performed non-trained finger sequences. In addition, classification accuracy increased for non-trained movements when using an increasing number of classifiers (Figure 2, Figure 5). The authors need to stress this important finding in the paper. Please provide an example for the force traces of a non-trained sequence in Figure 2 (as you did for a trained sequence in Figure 2) and mention this finding in the Abstract. The authors should also present maps showing how well learned and non-learned sequences were presented in the cortex at baseline and after training. This should be possible with their surface-based searchlight procedure (four maps) to visualize the general and sequence-specific effects of learning on the representations.

6) The sequence was cued by presenting a series of numbers. The pre-SMA has been shown to be critically involved in word repetition. This might have been a strategy to rehearse the sequence, especially for non-learned sequences and, thus, might have contributed to differences in the multivariate pattern associated with learned and non-learned sequences. This issue needs to be at least discussed in the revised paper.

7) An important result that should also be mentioned in the Abstract is that when decomposing the activity, 99% of the voxel-by-voxel variance was explained by a pattern component that is common to all trained sequences, with less than 1% being attributable to the sequence-specific component (Figure 6). This means that the sequence-specific pattern expression was much weaker than the patterns shared across sequences. This should be emphasized more in the Abstract and Discussion. This common component seems to increase with learning. Does the individual increase in explained voxel-by-voxel variance predict/correlate with the individual behavioral improvement of non-learned sequences.

8) The authors present some exciting and, in some cases, unexpected observations. For instance, while it is clear that sequence-specific decoding should emerge in well-documented areas like SMA or M1, the authors also report robust decoding in very posterior regions of the IPS as well as OPJ. This seems like a very important and novel finding as sequences have not been typically explored there (at least in the monkey neurophysiological literature). Yet, we could not find any discussion of these results in the paper. In addition, areas like PMd and even S1 show a set of interesting effects and these findings are not mentioned in the Discussion section. We believe that interesting questions can be asked of these regions. For instance, what type of “learning” do the authors think is taking place in S1 versus areas like OPJ? Given that no strict word limits are imposed by *eLife*, we would recommend that the authors seriously expand the scope of their Discussion to address many of these more novel findings.

---

## [Author Response]

*1) The authors acquired fMRI data for the learned and non-learned sequences in separate runs, and only apply classification within a fMRI run to test whether one learned sequence could be classified better against other learned sequences or whether one non-learned sequence could be classified better against other non-learned sequences (within-class classification). The authors need to comment on whether their data would also allow classification of learned sequences against non-learned sequences (comparison between classes of sequences as defined by the learning history) and to classify a learned sequence against other non-learned sequences. If such comparisons are not possible using the present dataset, please speculate on what one might expect*.

If we apply a classifier to distinguish between classes of sequences (trained vs. untrained), we obtain nearly perfect (100%) classification accuracies. However, this result is hard to interpret, as the trained and untrained sequences were measured in different runs and sessions, and any non-specific differences between the sessions (i.e., slight misalignment of the scans) could contribute to successful classification. We agree, however, that it would be desirable for future experiments to mix trained and untrained movements in a single run to be able to map all sequences into a single space.

The reviewers’ comment goes to the heart of the debate of what multivariate analysis is and how it should be used; we would therefore like to again clarify our approach here. If we employ multivariate techniques simply to compare two conditions (here trained and untrained sequences), each consisting of a number of different stimuli (sequences), we are basically asking the same scientific question as in the standard mass-univariate analysis: where are there differences in activation depending on training? In this case multivariate analysis is (as indeed often stated in the literature) simply a more sensitive way to detect differences that – if they were bigger and more systematic – could be theoretically picked up by standard mass-univariate analysis.

Here we decided on an experimental design in which we use a multivariate technique to determine how different stimuli (i.e., sequences) are represented within each condition (trained vs. untrained), and then compare the representational structure (classification accuracy, dimensionality of representation) across the two conditions. Thus we are here taking a “representational” rather than an “activation” approach.

*2) The authors could do a better job of weaving some key methodological details throughout their Results section. The authors begin discussing their behavioural results (Figure 2) and basic fMRI findings (in Figure 3) before a reader is made aware of the exact task that was employed. Figure 1 requires more details for a reader to fully understand the details of the task and appreciate the results. The authors should re-write parts of the beginning of the Results section to better explain to the reader the key components of the task*.

We have now reformulated the Results section to better explain methodological details of the behavioural task and imaging methods.

*3) A related problem is that fMRI activity was averaged over the cueing and execution phase. Would the same classification results emerge if classification were applied to the instruction phase and execution phase separately? It would be interesting to present classification results based on the cueing or execution phase separately*.

The current design is relatively fast; each trial consists only of a short (2.7s) instruction phase and a long execution phase (3 repetitions of the sequence – each triggered by a separate go signal – totaling 10.8s). This makes a reliable separation of instruction- and execution-related activity difficult. However, to address this issue, we modeled the response to each trial using a finite-response function (see Materials and methods). That is, one regressor modeled the activity for the first image after an instruction phase, the next regressor the second image, etc. We then submitted the beta values for each time point separately to a classification analysis. The results are now presented in Figure 4, which show the average activity and classification accuracy for different time points in a trial for the 6 bilateral ROIs. To compare the time courses, we normalized each time-series by its L2-norm (sums-of-squares).

The time x region interaction was significant for the normalized mean activity, F(40,600)=2.583, p<.001. Post-hoc tests of pairwise comparisons revealed that this difference arose because the two parietal regions (OPJ + IPS) were activated more strongly early in the trial, while the execution-related regions showed sustained activation during the execution phase.

For the classification accuracy, we found that that during the first time bin (TR = 0) no significant classification accuracy was reached. However, already during the second time bin (TR = 1, which was acquired 2.7s–5.4s after the onset of the imperative cue), the cross-validated accuracy was significantly above chance for S1, t(15) = 3.335, p = 0.002. This result indicates that some of the classification accuracy in this region may be driven by preparatory activity. For TR=2 (acquired 2.7–5.4 s after the onset of the first movement) all regions showed highly significant classification accuracy (all t(15)>5.34, p<.0001). However, activity in this image could already partly contain execution-related activity.

For the normalized accuracy we also found a significant time x region interaction, F(40,600)=1.67, p=.007. Post-hoc comparisons revealed that this interaction was due to the fact that OPJ and M1/S1 showed significantly different profiles: classification accuracy in OPJ peaked late, while it was already very high in early M1 and S1. The time courses of classification accuracy for trained and untrained sequences did not differ from each other.

In sum, these analyses indicate that the informative part of the activity patterns may have been related to preparatory activity, but certainly shows that the informative pattern is sustained in all regions across the whole execution phase. While the slightly different time-courses of classification accuracy are suggestive, we would like to be cautious in interpreting these, as the experiment was not designed to separate preparation and execution related activity. We therefore only stress the main insights from the analysis in the Results section, but do not discuss the inter-region differences. It will be interesting to explore the temporal dynamics of these patterns using a more widely spaced design.

*4) Figure 3 should include direct statistical comparisons between trained and untrained data in addition to the indirect comparisons provided in the overlap maps. Indirect comparisons are hard to interpret because “a difference in significance doesn’t necessarily indicate a significant difference”. The manuscript text describes the fact that trained sequences evoke less activity than untrained ones but the direct contrast, so it would be nice to see this in this Figure 3 and with p values (not just %SC values)*.

To allow for a comparison of the activity and classification accuracy between trained and untrained sequences, we chose to display the mean values rather than t-values in Figure 3. We believe that these overlay maps allow for the most direct comparison of difference and similarities of the two maps. The direct statistical contrasts were so far only reported in Table 2 and in the text. We now present the statistical maps for the direct contrast (t-values) in Figure 3. Finally, we also supplied a figure supplement to Figure 3, which presents separate maps for trained and untrained sequences in a color-coded scheme.

*5) The authors also find considerable general training effects when subjects performed non-trained finger sequences. In addition, classification accuracy increased for non-trained movements when using an increasing number of classifiers (Figure 2, Figure 5). The authors need to stress this important finding in the paper. Please provide an example for the force traces of a non-trained sequence in Figure 2 (as you did for a trained sequence in Figure 2) and mention this finding in the Abstract. The authors should also present maps showing how well learned and non-learned sequences were presented in the cortex at baseline and after training. This should be possible with their surface-based searchlight procedure (four maps) to visualize the general and sequence-specific effects of learning on the representations*.

We have now highlighted better in the Abstract, Results, and Discussion that untrained sequences can be well discriminated and appear to also be supported by a 3-dimensional representation (i.e., each sequence is associated with a unique activity patterns). We agree that this is an important finding and we have discussed possible sources of such representation in the Discussion.

We imaged the participants only after training, so we cannot compare the post-training classification performance to the baseline. An important motivation for this particular approach was that performance variability, error-rates, and movement times are usually much higher before training, making strong conclusions about learning-related changes difficult. We opted here for an approach that allowed us to determine the neural correlates of the sequence-specific part of sequence representations comparing trained to untrained sequences under well-controlled behavioral conditions. However, a more comprehensive study that looks at the development of sequence representations over the time-course of learning would be an important extension of the current work.

*6) The sequence was cued by presenting a series of numbers. The pre-SMA has been shown to be critically involved in word repetition. This might have been a strategy to rehearse the sequence, especially for non-learned sequences and, thus, might have contributed to differences in the multivariate pattern associated with learned and non-learned sequences. This issue needs to be at least discussed in the revised paper*.

Our current results only show tuning for sequential nature of the sequence, but do not reveal the exact nature of the representation in each of the areas. Thus, it is very well possible that activity patterns in pre-SMA were caused by subvocal rehearsal (19). We now mention this possibility, among others, in the Discussion. However, the fact that encoding was stronger for trained than untrained sequences makes a subvocal rehearsal explanation less likely, as participants (informally) reported that they relied less on verbal strategies for the trained compared to the untrained sequences.

*7) An important result that should also be mentioned in the Abstract is that when decomposing the activity, 99% of the voxel-by-voxel variance was explained by a pattern component that is common to all trained sequences, with less than 1% being attributable to the sequence-specific component (Figure 6). This means that the sequence-specific pattern expression was much weaker than the patterns shared across sequences. This should be emphasized more in the Abstract and Discussion. This common component seems to increase with learning. Does the individual increase in explained voxel-by-voxel variance predict/correlate with the individual behavioral improvement of non-learned sequences*.

The common component (which is strongly correlated with the level of average activity) actually was lower for the trained than untrained sequences. As the level of overall activity, the common component was not significantly related to amount of learning or the final level of performance.

*8) The authors present some exciting and, in some cases, unexpected observations. For instance, while it is clear that sequence-specific decoding should emerge in well-documented areas like SMA or M1, the authors also report robust decoding in very posterior regions of the IPS as well as OPJ. This seems like a very important and novel finding as sequences have not been typically explored there (at least in the monkey neurophysiological literature). Yet, we could not find any discussion of these results in the paper. In addition, areas like PMd and even S1 show a set of interesting effects and these findings are not mentioned in the Discussion section. We believe that interesting questions can be asked of these regions. For instance, what type of “learning” do the authors think is taking place in S1 versus areas like OPJ? Given that no strict word limits are imposed by* eLife*, we would recommend that the authors seriously expand the scope of their Discussion to address many of these more novel findings*.

We have now highlighted these findings and their implications in the Discussion. While the current results cannot reveal differences between sequence representations across primary and secondary motor areas, we are hopeful that the present investigation will provide the foundation for more detailed studies of the nature of these sequence representations. Indeed, in a follow-up study (in preparation) we were able to show that PMd and OPJ code sequences in an extrinsic (spatial) references frame, while IPS, M1, and S1 code the sequences in an intrinsic (joint or muscle based) reference frame. Given that these findings are beyond the scope of the current investigation, we have restricted our discussion to simply mention possible representational differences.